# A New Human Platelet Lysate for Mesenchymal Stem Cell Production Compliant with Good Manufacturing Practice Conditions

**DOI:** 10.3390/ijms23063234

**Published:** 2022-03-17

**Authors:** Katia Mareschi, Elena Marini, Alessia Giovanna Santa Banche Niclot, Marta Barone, Giuseppe Pinnetta, Aloe Adamini, Manuela Spadea, Luciana Labanca, Graziella Lucania, Ivana Ferrero, Franca Fagioli

**Affiliations:** 1Department of Public Health and Paediatrics, The University of Turin, Piazza Polonia 94, 10126 Torino, Italy; elena.marini@edu.unito.it (E.M.); alessiagiovannasanta.bancheniclot@unito.it (A.G.S.B.N.); marta.barone@edu.unito.it (M.B.); manuela.spadea@unito.it (M.S.); franca.fagioli@unito.it (F.F.); 2Stem Cell Transplantation and Cellular Therapy Laboratory, Paediatric Onco-Haematology Division, Regina Margherita Children’s Hospital, City of Health and Science of Turin, 10126 Torino, Italy; gpinnetta@cittadellasalute.to.it (G.P.); aloe.adamini@unito.it (A.A.); iferrero@cittadellasalute.to.it (I.F.); 3Blood Component Production and Validation Center, City of Health and Science of Turin, S. Anna Hospital, 10126 Turin, Italy; llabanca@cittadellasalute.to.it (L.L.); glucania@cittadellasalute.to.it (G.L.)

**Keywords:** mesenchymal stem cells, human platelet lysate, GMP

## Abstract

Mesenchymal stem cells (MSCs) are classified as advanced therapy medicinal products, a new category of GMP (good manufacturing practice)-compliant medicines for clinical use. We isolated MSCs from 5 bone marrow (BM) samples using human platelet lysate (HPL) instead of foetal bovine serum (FBS). We used a new method of HPL production consisting of treating platelet (PLTs) pools with Ca-Gluconate to form a gel clot, then mechanically squeezing to release growth factors. We compared the new HPL (HPL-S) with the standard (HPL-E) obtained by freezing/thawing cycles and by adding heparin. HPL-S had not PLTs and fibrinogen but the quantity of proteins and growth factors was comparable to HPL-E. Therefore, HPL-S needed fewer production steps to be in compliance with GMP conditions. The number of colonies forming unit-fibroblasts (CFU-F) and the maintenance of stem markers showed no significant differences between MSCs with HPL-E and HPL-S. The cumulative population doubling was higher in MSCs with HPL-E in the earlier passages, but we observed an inverted trend of cell growth at the fourth passage. Immunophenotypic analysis showed a significant lower expression of HLA-DR in the MSCs with HPL-S (1.30%) than HPL-E (14.10%). In conclusion, we demonstrated that HPL-S is an effective alternative for MSC production under GMP conditions.

## 1. Introduction

Medicine and biology have made many advances in cell therapy, and studies have shown that mesenchymal stem cells (MSCs) are ideal actors in the field of regenerative medicine and in the treatment of chronic degenerative diseases [1,2,3,4]. MSCs represent a very small percentage of the cells of the whole organism but they are distributed in almost all body organs [5]. However, bone marrow is the main source from which it is easy to isolate MSCs [6]. The minimal criteria for defining MSCs is outlined by the International Society for Cellular Therapy (ISCT) [7], who they define MSCs as multipotent adult stem cells present in various tissues, such as bone marrow, umbilical cord and fat tissue. They are characterized on the basis of their capability for self-renewing, adhesion and differentiation in cell lines such as osteoblasts, adipocytes and chondroblasts in vitro. As MSCs are multipotent cells, they are widely used in the treatment of various diseases due to their self-renewal, differentiation and immuno-modulatory properties through the release of exosomes and micro-vesicles containing numerous cytokines and growth factors [8]. The immunomodulatory properties of MSCs have been investigated extensively, although nowadays the exact mechanism of actions are not well-defined. Some studies have also shown the ability of MSCs to absorb huge amounts of chemotherapy drugs and to release them in a diseased area with exosomes [9].

The use of cells or tissues in cell therapy has led to the definition of a new category of medicines applied to the treatment of acquired or hereditary diseases: the advanced therapy medicinal products (ATMP), defined as preparations in which the main biological action is carried out by cells or tissues.

The production of ATMPs for clinical trials is subject to rigorous controls and regulatory requirements, which define the quality and safety criteria of the medicinal products and require their production in controlled and accredited institutions and agencies, in Italy, for example, by the Italian Medicines Agency (AIFA). Quality and safety must be guaranteed at each step of the production process: donation, isolation, expansion, quality control of manufacturing, conservation, storage and distribution. Good manufacturing practices (GMP) are the guidelines that describe the quality requirements for the production, quality control and release of a pharmaceutical product that also includes ATMPs. The purpose of GMPs is to ensure that a drug is produced, controlled and released by a certified quality system, in order to minimize the risks for patients [10].

GMP compliance is required by the European Directive 2003/24/EC, by the Italian Legislative Decree 219/2006, which acknowledges the indications, and by the European Regulation 1394/2007/CE. Significantly, Regulation 1394 establishes that ATMPs, including gene or cell therapies, are considered pharmaceutical products and are therefore subject to the guidelines governing GMPs. In addition, the European Directive 2001/20/EC and the Italian Legislative Decree 211/2003 also require the application of GMPs for pharmaceutical products undergoing clinical trial [11,12,13].

Process validation represents an important phase in the development of a medicine, in which the use of culture medium, additives and materials that can compromise the safety of medicines should be taken in account. The use of animal additives, for example, increases the risk of transmission of pathogens, which must clearly be considered in a careful risk analysis and in evaluating the use of alternative additives. The addition of human components to the culture medium instead of animal derivatives makes the cell expansion process compliant with GMP guidelines. To avoid using any animal products and to have a safe human product, we validated our MSC production in GMP conditions using human platelet lysate (HPL) inactivated with psoralen or riboflavin [14] as a substitute for foetal bovine serum (FBS), and we demonstrated how they preserve their multipotent and immunomodulant capacity [15]. HPL is rich in growth factors, cytokines and plasma proteins obtained from a pool of donor platelets (PLT) and also undergoes an inactivation process through the addition of psoralen or riboflavin to eliminate the possible presence of viral or bacterial nucleic acids. The standard production of HPL (HPL-E) consists of repeated freezing and thawing cycles of the platelet pool in order to release the growth factors contained within their granules and the addition of heparin to avoid the formation of gel in the culture medium [16,17]. The culture medium still requires numerous filtrations because, despite the addition of heparin, the formation of aggregates and debris deriving from HPL has been noted. However, these additional steps can pose a problem during extensive cell expansion under GMP conditions. Therefore, an alternative method for the production of HPL (HPL-S), which consists of making platelets coagulate through the addition of Ca-Gluconate and the subsequent mechanical wringing of the clot without the addition of heparin, could be fundamental for any extensive production of MSCs under GMP conditions. The result of this new method would be a much more limpid product than the standard and therefore the culture medium would not require numerous filtrations, thus eliminating many additional steps. A further advantage is that we are able to improve the safety of cell manipulation in GMP condition. The aim of this work was to verify if the new GMP method of HPL-S production is effective and preserves the characteristics of BM-MSCs when isolated from bone marrow (BM-MSCs).

## 2. Results and Discussions

This work investigated the effect HPL-S had on BM-MSCs when compared to HPL-E, which we considered standard HPL after having validated it as a good substitute for FBS in BM-MSCs isolation and expansion [14,15]. We performed a comparison between HPL-E and S using bone marrow sample, which is considered the main source of MSCs in our expertise. We investigated if the new HPL-S could be used for cell culture without the addition of heparin and filtration. This treatment could be very useful to avoid potential risks for patients during the manipulation of the cells, because this allows for the elimination of any animal-derived products and less manipulation under GMP conditions. We isolated MCSs from the waste bags of BM-related donors. All the BM donors were young and healthy as shown in Table 1.

First, we investigated if the treatment of Ca-Gluconate could interfere with the chemical characteristic and the growth factor release, then if the new HPL was effective in supporting the growth of MSCs.

### 2.1. HPL Production and Analysis

HPL was produced by the same pool of PLTs split into 2 bags and frozen at −30 C°. After thawing, the HPL-E appeared with debris and fragments that could have been platelet fragments, whereas the HPL-S appeared limpid and without debris, as shown in Figure 1A,C. In HPL-E, we added 200 IU/mL of heparin, and then both the HPL bags were divided into small aliquots and stored at −20 °C, ready to be used after thawing. Although the medium with HPL-S did not require filtration, we routinely filtered it to treat the two HPLs in the same way at the moment of the medium preparation. Although we observed the presence of debris and aggregates after filtration in the medium, as shown in Figure 1B,E, and then also in the culture observing the cells at the microscope during the expansion, as illustrated in Figure 1C,F.

#### 2.1.1. Biochemical Analysis

The HPL batches were prepared from a pool of platelets from 10 donors. Their chemical analysis revealed that the mean of PLT present in the samples was 1037.25 (×10^3^/μL). After standard treatment in the HPL-E, a residual PLT concentration persisted, because the PLT concentration was a mean of 72.50 with a SEM of 14.40 × 103/μL, while in HPL-S, PLT was almost completely absent (mean value with SEM was 0.75 ± 0.75 × 10^3^/μL), as seen in Figure 2A. To verify if the treatment of Ca-Gluconate could interfere with the protein content, an evaluation of total proteins was performed, and we obtained a mean of 5.35 ± 0.75 g/dL and 5.025 ± 0.53 g/dL, respectively, in HPL-E and HPL-S, without statistically significant differences (Figure 2B). The absence of the fibrinogen in the HPL-S confirmed the effect of the treatment with Ca-Gluconate, while, in the HPL, a significant presence persisted (80.50 ± 10.65 mg/dL), as shown in Figure 2C.

The absence of fibrinogen in the HPL-S was related to their manipulation performed during the wringing of the PTL lysate that eliminated much more PLT residue. These results demonstrated an advantage of HPL-S versus the standard HPL for the following reasons:(1)The HPL-S was more limpid, and no debris was present.(2)No heparin was added to avoid the coagulation of the medium through the clumping of the fibrinogen in the HPL.(3)No filtration was needed for the culture media, and therefore no additional manipulation occurred under GMP conditions.

#### 2.1.2. Growth Factor Analysis

To verify if the treatment with Ca-Gluconate could also interfere with the release of growth factors, ELISA assay was performed on both HPLs. The relation coefficient R2 value of the standard curve was more than 0.96 for each analysed growth factor. In one batch of both HPL, we found some non-detectable results for EGF and FGF. The results for EFG, VEGF, FGF, PDGF were reported as a mean with SEM in Figure 3. We observed a light decrease of protein quantity and the release of growth factors such as EGF, VEGF, PDGF, FGF, but overall, no statistically significant differences were reported, demonstrating that the treatment with Ca-Gluconate did not modify the release of proteins and growth factors. These results showed that HPL-S can be used instead of HPL-E for the isolation and expansion of BM-MSCs under GMP conditions. In addition, IFN-γ was researched, but no release was detected in both HPLs.

### 2.2. Sample Collection and Colony-Forming Unit Fibloblastoids

We verified if HPL-S was able to support the growth of MSCs on a large scale by comparing the new HPL with standard HPL (HPL-E) in the five production batches of BM-MSCs.

The MSCs were isolated from the BM collection of healthy donors with a median age of 23 years (range: 20–48 years). Details of donors’ characteristics are described in Table 1.

Adherent fibroblastoid colonies were observed in all of the batches with fibroblastic morphology after 7 days of culture, as shown in Figure 4A,B. The appearance of the colonies showed a higher concentration of cells in HPL-E than in HPL-S, without differences in morphology (Figure 4A,B). Although the number of CFU-F was always higher in BM-MSCs cultivated in HPL-E compared to BM-MSCs in HPL-S, a paired *t*-test did not show any statistically significant differences. The mean with SEM of CFU-F/10^6^ cells was 127.00 ± 37.93 and 92.00 ± 32.47 in BM-MSCs cultivated, respectively, in HPL-E and in HPL-S (Figure 4C), without statistical differences.

### 2.3. Cellular Growth

Because the clonogenic potential is extremely variable for each donor, we also considered cell growth expansion.

Cells at confluence were detached after a mean of 16.80 days (with a SEM of 1.241) at the first passage and then at the second, third and fourth passages after 14.40 days ± 2.62, 11.00 ± 0.55 and 20.67 ± 0.33, respectively. Although a lower trend was observed in BM-MSCs cultivated in HPL-S, with an inversion of the trend in the last passage, both conditions led them to expand significantly over time without statistically significant differences, as demonstrated by ANOVA in multi-comparison analyses (*p* > 0.05) and as shown in Figure 5. The cell proliferative capacity during expansion was expressed as population doubling (PD) using the following formula: Log_10_N/Log_10_2, where N was the cell number of the detached cells divided by the initial number of seeded cells. The mean of errors of cumulative PD obtained at each passage are summarized in Table 2.

In GMP conditions, MSC production usually is not performed past the third passage [18], but we continued the expansion until the fifth passage to evaluate if HPL-S supports growth over time. No more passages were needed for our aim.

### 2.4. Immunophenoipic Analysis

To verify that expanded cells, in both HPLs, were MSCs, we performed an immunophenotype at p2 as suggested by the International Society for Cell and Gene Therapy for defining multipotent mesenchymal stromal cells [7]. As seen in Figure 6, the immunophenotype analysis showed positivity for CD105, CD73 and CD90 and negativity for CD45-34–14 and CD19 in cells cultured with HPL-E and HPL-S, without any differences. We noticed a very significant difference (*p* = 0.015) in the expression of CD146 that is higher in MSCs treated with HPL-S (>50%) with respect to those treated with HPL-E (<30%). Moreover, we observed a significant difference for the expression level of HLA-DR in HPL-E, with an expression of 17.27% ± 5.91 compared to 0.54% ± 0.16 of those cultured with HPL-S. Contrary to the minimal criteria proposed by the International Society for Cell and Gene Therapy for defining multipotent mesenchymal stromal cells (MSC), HLA-DR expression was analysed and used only for informative purposes but not for product release because it is largely unpredictable in ex vivo-expanded clinical-grade cultures [19]. In previous studies, we also observed a high expression of HLA-DR in MSCs cultivated in standard HPL, and, because IFN-γ could interfere with the expression of HLA-DR, we tested the release of IFN-γ in the 2 HPLs with ELISA, which resulted in the complete absence of IFN-γ. For this reason, we hypothesize that the presence of some residue PLTs could be responsible for the presence of HLA-DR on the MSCs that they do not normally express.

The absence of HLA-DR in BM-MSCs cultivated in HPL-S guarantees their compliant characterization as MSCs, giving them a higher immunological privilege for allogenic clinical use. On the other hand, we saw a very significant difference in the expression level for CD146. Those cells, cultured with HPL-S data, demonstrated a higher level of CD146. Recently, Bowles et al. [20] demonstrated that CD146 + BM-MSCs had a markedly higher secretory capacity with significantly greater immunomodulatory and anti-inflammatory protein production upon inflammatory priming compared with the CD146- BM-MSCs. For this reason, we will investigate the immunomodulant properties of MSCs isolated in the two HPLs.

### 2.5. Differentiation Capacity

We also analyse the multipotent capacity of BM-MSCs isolated in the two HPLs. After induction with specific culture medium, we were able to differentiate all batches of BM-MSCs in osteoblasts, adipocytes and chondrocytes, as shown in a representative experiment in Figure 7. No differences were identified in our experiments.

### 2.6. Stem Cell Marker Expression

To determine if MSCs cultivated in the 2 HPLs maintained their stemness through each passage, from p0 to p4, we evaluated the expression level of NANOG, OCT-4, SOX-2 and stem cell markers. A real-time PCR was performed on five MSC samples. No significant differences between BM-MSCs cultivated in HPL-E and HPL-S were observed in gene expression at any cell passage (Figure 8), meaning that both cell lines displayed similar stemness phenotype that was steadily maintained during the in vitro culture.

Because various protocols for HPL generation are available from the literature [16], the International Society of Blood Transfusion (ISBT) has recently focused on HPL production methods and proposed recommendations on manufacturing and quality management in line with current technological innovations and regulations of biological products and advanced therapy medicinal products [21]. Our data confirmed that HPL-S could be a valid substitute for HPL-E and a new method to validate MSC production in GMP conditions.

## 3. Materials and Methods

### 3.1. HPL Production and Analysis

#### 3.1.1. HPL Production

Each HPL batch was prepared from two buffy-coat derived platelet concentrates (BC-PCs).

In detail, five O-group BCs were pooled with one AB-group plasma, then centrifuged and automatically separated through a leukoreduction filter (TACSI system, Terumo BCT Europe, N.V. Zaventem, Belgium) to obtain the platelet concentrate (PC) (pooling of 10 units of platelet concentrates from buffy coat (BC) pools. Each PC was prepared by pooling 5 units of BC group 0 and one unit of plasma group AB).

The BC-PCs units were then inactivated by the Mirasol PRT system (Terumo BCT, Lakewood, CO, USA). Briefly, after the addition of 35 mL of riboflavin solution (500 μM riboflavin in 0.9% sodium chloride solution), the BC-PC unit was placed into the illuminator and exposed to UV light (6.24 J/mL) with continuous horizontal shaking.

Following the treatment procedure, two BC-PC units were pooled and split into two aliquots for the production of HPL-E and HPL-S. Samples were taken from each unit to perform a platelet count (HPL pre).

The HPL-E was produced by platelet lysis, induced by three freeze/thaw cycles (−30 °C/37 °C) and platelet fragments depletion by centrifugation (5300× *g*, 8 min). Samples were taken before cryopreservation and stored at −30 °C for platelet count and sterility testing.

The HPL-S was produced by treating platelets with 20 mL of Ca-gluconate (1000 mg/10 mL) and heating the bag unit to 37 °C until clots formed. The coagulated medium was then centrifuged (5300× *g*, 8 min) and squeezed to recover a clear supernatant that only contained growth factors without platelet fragments. Samples were taken before cryopreservation and stored at −30 °C for platelet count and sterility testing.

#### 3.1.2. Biochemical Analysis

Platelets were counted in Sysmex XE2100.

A quantitative determination of total proteins was performed by colorimetric test with a Roche/Hitachi cobas c system.

Plasma coagulation factors assays were carried out using the automated coagulation analyser ACL (Werfen, Instrumentation Laboratory, Barcelona, Spain).

Factor VIII activity was measured using the activated partial thromboplastin time (APTT) method and a factor-deficient substrate. The sample was combined and incubated with a factor VIII-deficient substrate (normal plasma depleted of factor VIII by immunoadsorption) and an APTT reagent. After an established incubation time period, calcium was added to trigger the coagulation process in the mixture. Then the time frame to clot formation was measured optically at a wavelength of 671 nm.

Fibrinogen concentration was measured by the Clauss fibrinogen assay.

The sample, containing fibrinogen, was mixed with a reagent containing excess thrombin. The excess thrombin converts the fibrinogen in the sample to fibrin. The amount of time it takes to form a clot is inversely proportional to the amount of fibrinogen present in the sample.

A fibrinogen reference curve was plotted from the clotting time results of the known reference plasma dilutions that expressed different fibrinogen values. The concentration of fibrinogen in plasma samples was determined by comparing clotting time values to the reference curve.

#### 3.1.3. Growth Factor Analysis

An aliquot of each HPL was collected for the evaluation of concentration levels of VEGF, PDGF, EGF and FGF.

IFN-γ was also analysed to evaluate if its presence can modulate HLA-DR, as shown in Guess et al. [22].

The analysis was performed using an ELISA kit (Invitrogen, ELISA Assay), following the manufacturer’s instructions.

Briefly, 50–100 µL of standards and samples were plated in duplicate on a well that was already coated with the antibody and incubated at room temperature (RT) for 2–3 h. Four washes were performed and then 50–100 µL of biotinylated antibody 1× for 1 h at RT were added and incubated for 1 h at RT. A further four washes were performed before adding streptavidin-horseradish peroxidase. At the end of the procedure, we washed it four times and incubated with an appropriate stabilized chromogen, conferring a colouring, on reaction, with an intensity (evaluated with a spectrophotometer at wavelength of 450 nm) directly proportional to the factor concentration. Data were expressed as the concentration in picograms per millilitre.

### 3.2. Sample Collection and Cell Cultures

Samples were obtained from the collection of bone marrow (BM) by washing the post-infusion waste bag. The samples were named BM-MSC-1, BM-MSC-2 and BM-MSC-3. Each sample was cultivated in two parallel cultures with Alpha Medium (Sigma Aldrich, St. Louis, MI, USA), 1% of L-glutammine, 1% of pen/streptomycin and 10% of HPL-E or HPL-S, respectively. During their expansion, the cells were kept in an incubator at 37 °C and 5% CO2. At the initial seeding the cells were plated at 1 × 10^4^ cells/cm^2^, and, 7 days later, the original medium with non-adhered cells was removed. A medium change was performed every 2–3 days. When the cells reached around 80% of confluence, they were detached using trypsin (trypsin/EDTA, Sigma-Aldrich, St. Louis, MI, USA) for 5 min at 37 °C in incubator. From the first step onwards, they were plated at a concentration of 1 × 10^3^ cells/cm^2^.

At each step, the cells were detached, and the following analyses were carried out: cell count and viability by staining with Trypan blue and counting in a Burker chamber, immunophenotype analysis by flow cytometry, mRNA extraction for subsequent real-time PCR and, only at the third step, differentiation induction. After setting up all the experiments, at each step, the excess on the detached cells was cryopreserved as a cell suspension in cryovials, in α-MEM 5% of human albumin (Kedrion, Lucca, Italy) and 10% of dimethyl sulfoxide (DMSO) (Sigma Aldrich, St. Louis).

### 3.3. Colony-Forming Unit Fibloblasts

To quantify MSC precursors, we performed a colony-forming unit fibroblast (CFU-F) test. The BM cells were seeded in six well-plates in α-MEM containing 10% HPL-E or HPL-S at 10,000. After 7–10 days, MSC clonogenic precursors were fixed in acetone methanol and coloured with the May Grunwald dye in order to proceed to the visual counting of the colonies. Clusters of >50 cells were considered colonies and scored macroscopically 7–10 days after seeding.

All experiments were performed in duplicate and by two different operators. The CFU-F means were expressed as fibroblastic clones obtained from 1 million BM cells (CFU-F/10^6^ cells).

### 3.4. Count and Cell Viability

The cells were analysed for cell growth by calculating population doubling (PD) with the following formula: Log_10_ N/Log_10_ 2, where N (growth rate) is the number of cells recovered divided by the initial number of cells seeded. Their expansion was expressed in terms of cumulative PD (cPD).

A vitality assessment was performed at each step during cell expansion.

The cells were counted at optical microscope, using Burker chamber calculation as indicated in the European Pharmacopoeia (Chapter 2.7.29).

### 3.5. Immunophenotype Analysis

At each step the characteristic MSC surface markers were analysed by flow cytometry using the following antibodies: CD90 FITC, CD73 PE, CD105 PC7, CD45-34-14 FITC, HLA-DR PE, CD19 APC and CD146 APC.

A total of 1 × 10^6^ cells were divided into 5 tubes in aliquots of 100 µL of cell suspension in buffered saline solution (PBS). The tubes were then incubated for 20 min at 4 °C with the addition of 10 µL of monoclonal antibody conjugated with a fluorochrome (fluorescein isothiocyanate: FITC, phycoerythrin: PE, allophicocyanine: APC) and were then analysed using Navios software (Vs. 1.2, Beckman Coulter, Krefeld, Germany). Non-labelled antibody cells were used as negative control.

### 3.6. Differentiation Induction

In order to investigate the multipotent capacity of the isolated BM-MSCs, we cultivated them in specific culture medium inducing osteoblastic, adipogenic and chondrogenic differentiation (Miltenyi Biotec, Bergisch Gladbach, Germany). The duration of the culture was 15 days for the osteogenic medium and 21 days for the adipogenic and chondrogenic media, according to the manufacturer’s instructions.

#### 3.6.1. Osteoblastic Differentiation

Four aliquots of 45 × 10^3^ cells were grown in 35 mm Petri dishes: two plates were set up with differentiating medium StemMACS OsteoDiff Media (Miltenyi, Germany), and the other two were used as control. To verify cell differentiation, after 15 days, differentiated BM-MSCs were fixed with acetone/methanol (1:1), stained with Van Kossa and observed morphologically using an optical microscope. The presence of calcium crystals and mineral deposits was evaluated.

#### 3.6.2. Adipogenic Differentiation

Four aliquots of 75 × 10^3^ cells were cultured in 35 mm Petri dishes: two plates were set up with StemMACS.

AdipoDiff Media differentiation medium (Miltenyi, Germany) and the other two were used as control. The changing of the medium was performed three times a week, and, after 21 days, cells were fixed with paraformaldehyde (PAF) vapours for 10 min and stained with oil red O dye. They were morphologically observed using an optical microscope to evaluate the presence of red lipid vacuoles, the differentiated and coloured cells having red lipid granules and blue nuclei.

#### 3.6.3. Chondrogenic Differentiation

Aliquots of 25 × 10^4^ cells were resuspended in tubes (it is handy to use 15 mL) in 1 mL of StemMACS ChondroDiff Media differentiation medium (Miltenyi, Germany) and were grown as a three-dimensional cell aggregation for 21 days by changing the culture medium three times a week. At the end of the third week the differentiation was evaluated by Alcian Blue staining.

### 3.7. Rna Extraction, Reverse Transcription and Real-Time PCR

The stemness of MSCs was evaluated by the expression of specific stemness markers: Homeobox protein (NANOG), octamer-binding transcription factor 4 (OCT4) and SRY (sex determining region Y)-box 2 (SOX2), which were analysed by real-time PCR. A total of 1 × 10^6^ MSCs expanded in the two types of HPL cells were detached at each passage and centrifuged at 13,200 rpm for 5 min in order to obtain a dry pellet. The pellets were then stored at −20 °C for subsequent total RNA extraction using Purelink RNA mini kit (Life Technologies Italia, Monza, Italy). The extracted RNA was quantified by QIAxpert (QI-AGEN, Hilden, Germany), and 200 ng of material was retrotranscribed with SSIV VILO MASTERMIX W/EZDNASE (Life Technologies Italia, Monza, Italy).

Their RNA was retro-transcribed into cDNA using GeneAmp, PCR System 9700 Thermal Cycle (Applied Biosystems) under the following conditions: 10 min at 25 °C, 10 min at 50 °C, 5 min at 85 °C.

The cDNAs obtained were stored at −80 °C until further amplification.

The relative expression quantification of the selected genes (obtained with Taqman En-zyme amplification process, StepOne, Real-time PCR System, Thermo Fisher Scientific, Waltham, MA, USA) was obtained by normalization with the Glyceraldehyde-3-Phosphate Dehydrogenase housekeeping gene (GAPDH Hs99999905_m1, Thermo Fisher Scientific, Waltham, MA, USA). The markers used were Nanog (Hs02387400_g1, Thermo Fisher Scientific, Waltham, MA, USA), Oct-4 (Hs03005111_g1, Thermo Fisher Scientific, Waltham, MA, USA) and Sox-2 (Hs01053049_s1, Thermo Fisher Scientific, Waltham, MA, USA) and 15 ng of cDNA were amplified in 10 μL of total reaction volume containing:–5 μL of Taqman fast advanced mastermix (4444556, Thermo Fisher Scientific, Waltham, MA, USA),–0.5 μL of Taqman Gene ex assays (4453320, Thermo Fisher Scientific, Waltham, MA, USA),–4.5 μL of cDNA,

The reaction mixes were deposited in 96-wells plates using the following conditions: 2 min of hold at 50 °C and 2 min of hold at 95 °C, followed by 40 cycles of 1 s at 95 °C and 20 s at 60 °C. Each sample was analyzed in triplicate. We compared the expression of the target gene in the different batches using ΔCt values calculating as [CT target gene—CT housekeeping gene] during their expansion.

### 3.8. Statistical Analysis

Statistical analyses were performed with the use of GraphPad Prism statistical software. The differences between paired samples were evaluated with *t*-test analysis. We considered a significant difference if their *p* value was <0.05.

Comparisons of HPL-E and HPL-S across different groups were done with an unpaired two tailed *t*-test or by two-way analyses of variance (ANOVA).

To compare the effect of HPL-E and HPL-S on different groups, a multi-comparison analysis two-way ANOVA was performed.

## 4. Conclusions

We can conclude that HPL-S could be a better cell culture supplement for MSC expansion in GMP conditions. The results obtained in our experiments suggest that we can introduce HPL-S as a valid alternative for HPL-E while avoiding the addition of heparin and any animal additive during the production process and furthermore avoiding the medium filtration step. This contributes to mitigating the risk of contamination during cell production.

Because our interest in cellular therapy with MSCs is now focused on the production of secretome and its clinical use, in this study, we have also collected the culture medium and isolated the secretome as described in Bari et al. [23]. We also analysed the differences in the secretome isolated from BM-MSCs and expanded in HPL-E and also those in HPL-S to investigate theirmmunomodulant effects on activated lymphocytes. Furthermore, we compared the effect of HPL-E and HPL-S on the MSC secretome, analysing their physical, chemical, immunophenotypic and functional characteristics. We observe that the secretome produced by MSCs, isolated and cultured with the two different HPLs, did not show significant characteristics and preserve the paracrine effect and immunomodulant proprieties on activated lymphocytes [24] (manuscript submitted). All of these data allow us to identify HPL obtained from a pool of PLTs that underwent Ca-Gluconate treatment, and the subsequent mechanical wringing of the clot is a good alternative to the standard HPL and FBS to isolate and expand MSCs on a large scale in GMP condition, preserving their immunophenotipic, multipotent and immunomodulant characteristics.

## Figures and Tables

**Figure 1 ijms-23-03234-f001:**
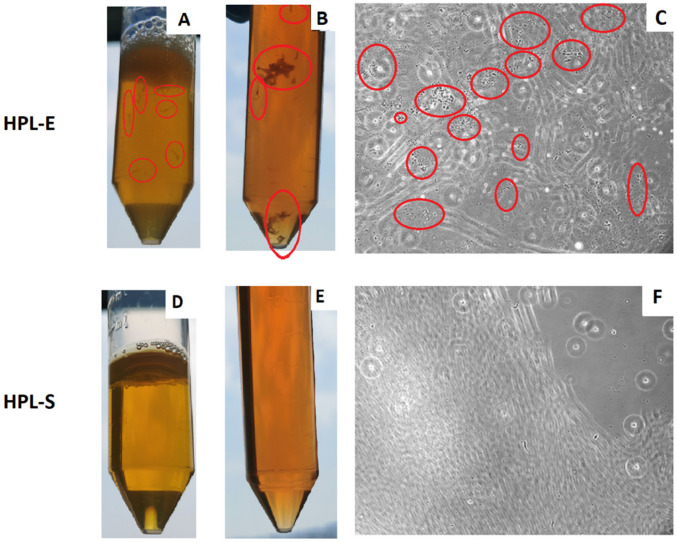
Pictures of HPL-E (**A**) or HPL-S (**D**) after thawing and of the medium Alpha MEM with HPL-E (**B**) or HPL-S (**E**) two days after the preparation and filtration, and representative microscopic observations of the MSCs in the BM-MSCs in HPL-E (**C**) and in HPL-S (**F**) during the culture (Magnification 10×), focusing on the debris. Red circles are placed on evidence of platelet aggregates and debris present only in the HPLE condition.

**Figure 2 ijms-23-03234-f002:**
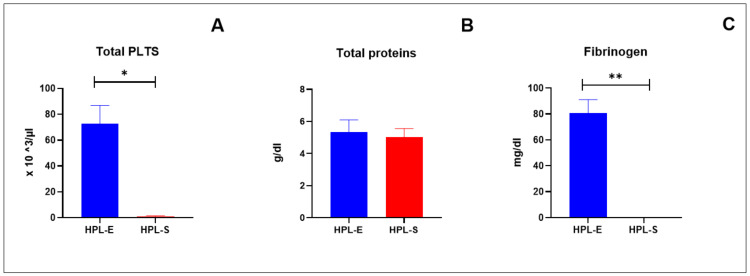
Chemical analyses performed in HPL-E (blue column) and in HPL-S (red column). Each column represents the mean with SEM of 4 batches of each HPL. No significant differences were observed between HPL-E and HPL-S in total proteins (**B**), but the absence of fibrinogen (**C**) and total PLTs (**A**) were significant in HPL-S in comparison with HPL-E. * and ** indicate, respectively, a significant (*p* < 0.05) and a highly significant (*p* < 0.01) difference.

**Figure 3 ijms-23-03234-f003:**
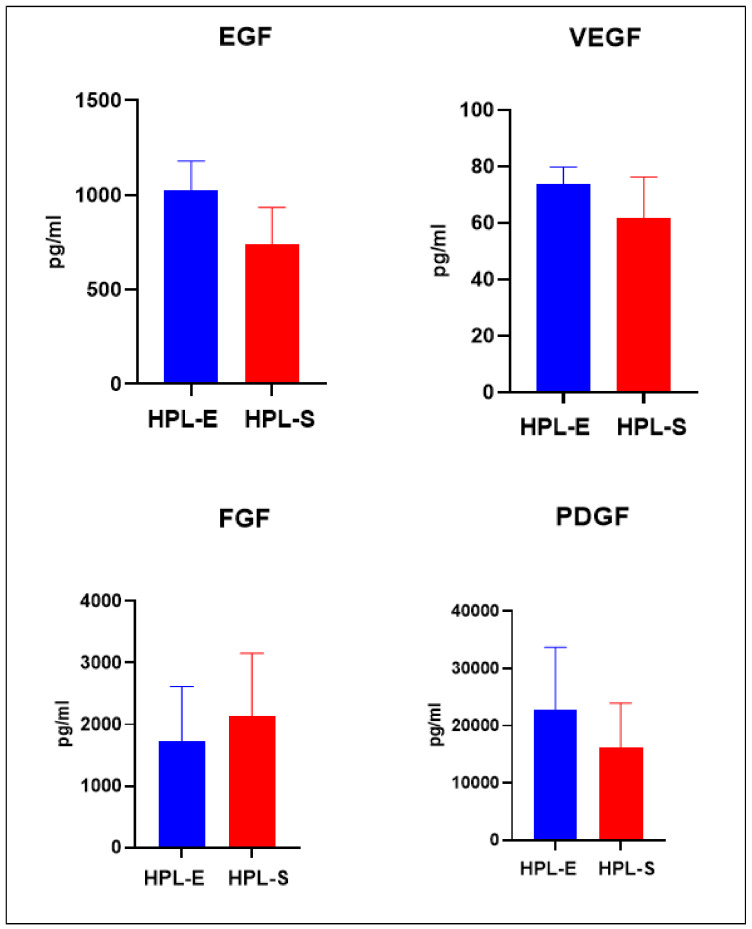
Growth factor release in HPL-E and HPL-S. Each column represents the means with SEM of 4 batches of each HPL. No significant differences were observed between HPL-E and HPL-S.

**Figure 4 ijms-23-03234-f004:**
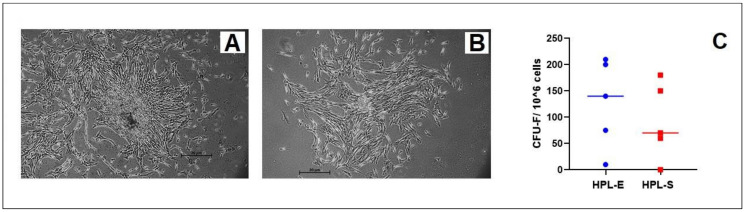
Representative phase images at 5× magnification of CFU-Fs after 7 days from seeding. CFU morphology of BM-MSCs cultivated in HPL-E (**A**) and in HPL-S (**B**). In (**C**), blue and red dots report CFU-F/10^6^ cells of BM-MSCs batches with HPL-E and HPL-S, respectively. No significant differences were observed.

**Figure 5 ijms-23-03234-f005:**
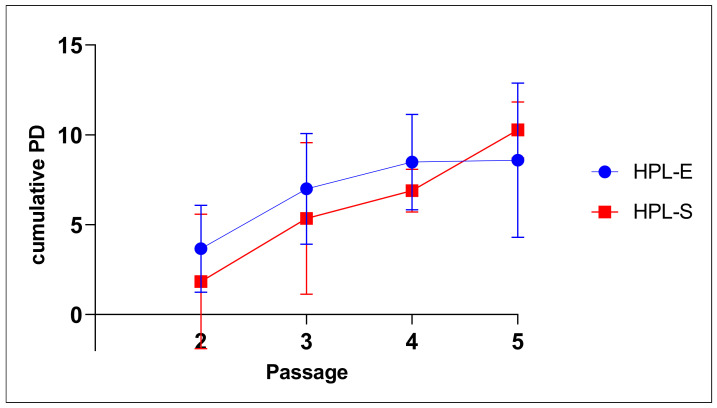
Cellular growth expressed as cumulative PD from 2nd to 5th passages during expansion on BM-MSC cultivated in HPL-E (blue curve) and HPL-S (red curve).

**Figure 6 ijms-23-03234-f006:**
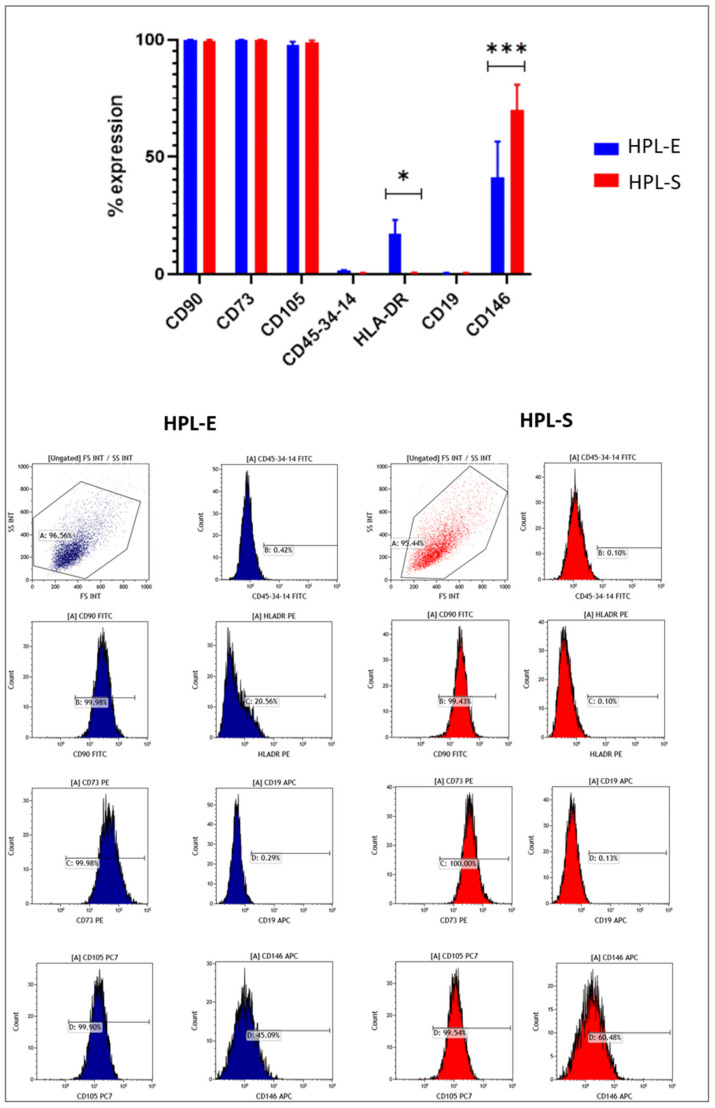
Immunophenotype analysis. Values are expressed as mean percentage ± SEM of positive cells for MSC antigen expression in histogram bars (top panel). Cells were analyzed at the 2nd passage for each experimental condition (n = 5), and a representative cytofluorimetric analysis is illustrated in the bottom panel. * and *** indicate, respectively, a significant (*p* < 0.05) and a very highly significant (*p* < 0.001) difference.

**Figure 7 ijms-23-03234-f007:**
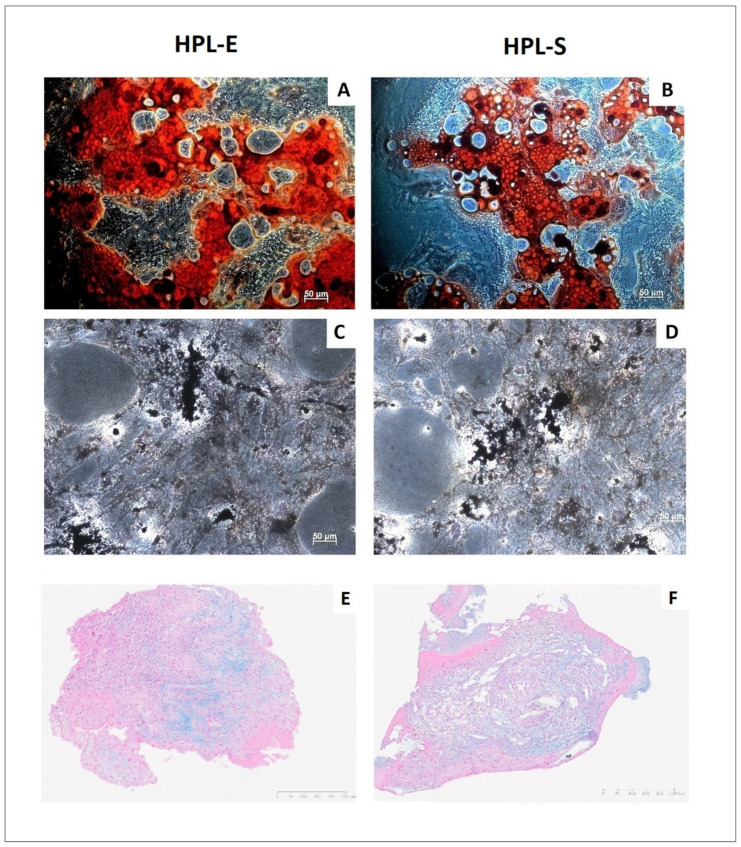
MSC differentiation potential assay after specific induction in MSCs. Representative images of oil red O staining (**A**,**B**) showing intracytoplasmic vacuoles in adipocytes, Von Kossa staining (**C**,**D**) showing the presence of calcium oxalates in osteoblasts, and Alcian blue (**E**,**F**) showing the hyaluronic acid for chondrocytes, respectively, in HPL-E (left images) and HPL-S (right images).

**Figure 8 ijms-23-03234-f008:**
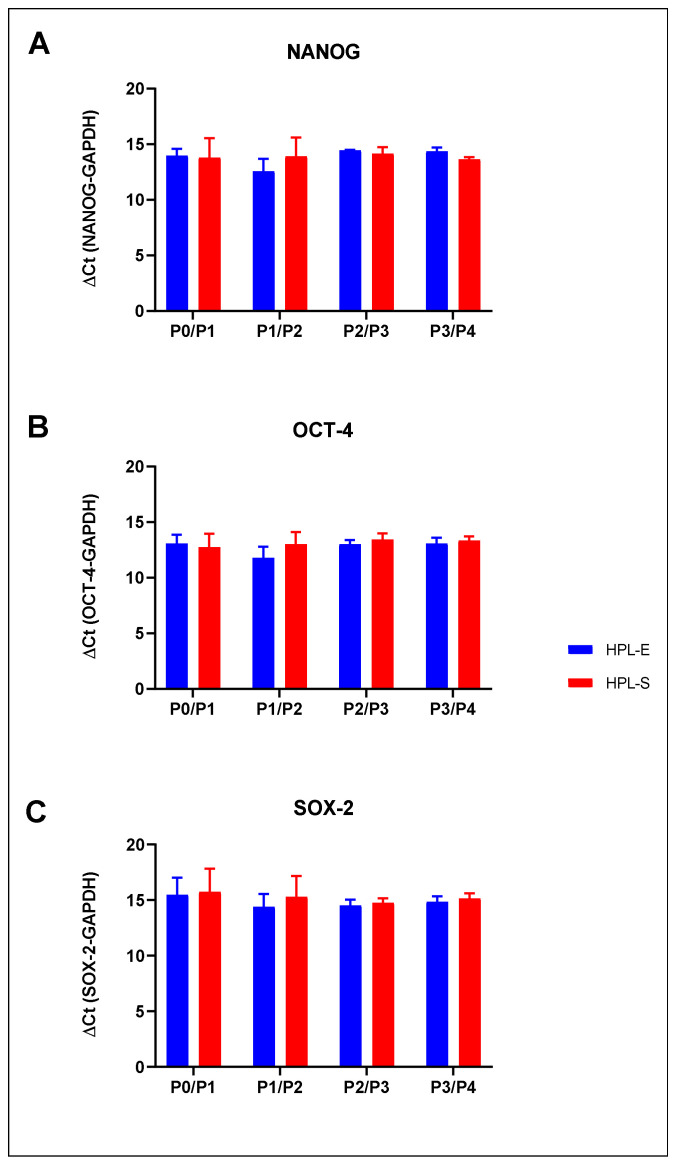
Stem cell markers analyses by RT-PCR: NANOG (**A**), Oct-3/4 (**B**) and Sox-2 (**C**) expression is shown as means with SEM of five experiments in HPL-E (blue columns) and HPL-S (red columns) at different passages. No significant differences were observed between the HPLs and between the passages.

**Table 1 ijms-23-03234-t001:** MSCs donors’ age and sex.

ID BM-MSC	Age	Sex
BM-MSC-01	20	Male
BM-MSC-02	23	Male
BM-MSC-03	48	Male
BM-MSC-04	22	Male
BM-MSC-05	40	Male

**Table 2 ijms-23-03234-t002:** Cumulative PD of BM-MSCs obtained at each passage during expansion in HPL-E and HPL-S. Data are expressed as means with the standard error of 5 experiments.

	HPL-E_p2	HPL-S_p2	HPL-E_p3	HPL-S_p3	HPL-E_p4	HPL-S_p4	HPL-E_p5	HPL-S_p5
Mean	3.66	1.84	6.1	5.35	8.49	6.89	8.59	10.27
Std. Error of Mean	1.08	1.68	1.38	1.87	1.53	0.67	2.48	0.90

## Data Availability

All data generated or analysed during this study are included in this published article. However, the data obtained in this study are also available from the corresponding author upon request.

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
