# Peer review of "A New Human Platelet Lysate for Mesenchymal Stem Cell Production Compliant with Good Manufacturing Practice Conditions"

_ijms, 2022, doi:10.3390/ijms23063234_

Round 1

Reviewer 1 Report

In this study, the authors propose a new alternative method for the production of human platelet lysate to use for mesenchymal stem cell culture and expansion. They use calcium gluconate to aggregate platelets following by a mechanic wringing process to obtain the final product. They claim this alternative method is easier and obtain a much more limpid lysate than the conventional ones avoiding the use of heparin and filtration process. Apart from platelet lysate analysis taking into account the presence of fibrinogen and different growth factors, they verify the effect of the new product on the mesenchymal stem cells in terms of colony-forming units, cell growth, phenotype and differentiation capacity.

Scientific background, rationale of the study and GMP compliance required for this type of studies as well as the research design and the presentation of the results are well described. The methodology they propose is very interesting with a high impact in the field of culture media for MSC culture and expansion. The findings are very interesting since nowadays the culture media for MSC is currently investigated to improve the product and decrease the costs.

Minor points

In general, check misspelled words

Abstract section;

GMP, should be defined

Introduction section;

-The sentence “The immunomodulatory properties of these cells have not been investigated extensively but this could potentially be a very innovative tool for cellular therapies” in page 2 is not completely right. This sentence should be edited. The immunomodulatory properties of MSCs have been investigated extensively although nowadays the exact mechanism of actions are not well defined.

-In the 4th paragraph, reference number 14 does not match

Result and Discussion

In general, t-test use should be explained

Chemical analysis section

-First mention to HPL-S and HPL-E should be defined

-Platelet donor information should be added

-Maybe “Chemical” word is not correct for this section

-Total protein units should be added

Sample collection and colony-forming unit fibloblast section

-They are not fibroblast. They have fibroblast morphology. Fibroblastoid colony-forming unit should be better

Cellular growth section

-First use of “PD” should be defined

-In table 2, Decimal numbers are written in an incorrect form

Differentiation Capacity

- "HPL-S" and "HPL-E" in the top of the images of figure 6 help the reader

- Poor resolution in Panel E and F of figure 6

Material and Methods

In Result section, 5 donors appear in contrast to the number appears in this section

Author Response

Dear reviewer, 

Please find enclosed the revised manuscript entitled “A New Human Platelet Lysate for Mesenchymal Stem Cell Production Compliant with Good Manufacturing Practice Conditions “ 

We wish to thank you for giving us the opportunity to revise and improve our work. 

In this revised version we took into consideration the reviewers’ comments and modified the text accordingly. All alterations in the revised manuscript are tracked using the      "Track Changes" function in Microsoft Word.

We believe that by following the reviewers’ suggestions the new version has been much improved, and hope that it is now suitable for publication in your journal. 

Please, contact us without hesitation  with regard to any further questions or modifications.  

Yours sincerely, 

Katia Mareschi

Below is our response to your comments. 

In this study, the authors propose a new alternative method for the production of human platelet lysate to use for mesenchymal stem cell culture and expansion. They use calcium gluconate to aggregate platelets following by a mechanic wringing process to obtain the final product. They claim this alternative method is easier and obtain a much more limpid lysate than the conventional ones avoiding the use of heparin and filtration process. Apart from platelet lysate analysis taking into account the presence of fibrinogen and different growth factors, they verify the effect of the new product on the mesenchymal stem cells in terms of colony-forming units, cell growth, phenotype and differentiation capacity.

Scientific background, rationale of the study and GMP compliance required for this type of studies as well as the research design and the presentation of the results are well described. The methodology they propose is very interesting with a high impact in the field of culture media for MSC culture and expansion. The findings are very interesting since nowadays the culture media for MSC is currently investigated to improve the product and decrease the costs.

We thank the reviewer for the thorough and positive assessment of our work. 

Minor points

In general, check misspelled words

An     English     mother-tongue has reviewed the whole manuscript.

Abstract section;

GMP, should be defined

We have defined it.

Introduction section;

-The sentence “The immunomodulatory properties of these cells have not been investigated extensively but this could potentially be a very innovative tool for cellular therapies” in page 2 is not completely right. This sentence should be edited. The immunomodulatory properties of MSCs have been investigated extensively although nowadays the exact mechanism of actions are not well defined.

We have edited the sentence as suggested.

-In the 4th paragraph, reference number 14 does not match

We have changed the references.

Result and Discussion

In general, t-test use should be explained

We added an explanation in the Material and method session

 Chemical analysis section

-First mention to HPL-S and HPL-E should be defined

We have defined HPL-E and HPL-S in the Introduction.

-Platelet donor information should be added

We have added this information in the M     aterial and method section

-Maybe “Chemical” word is not correct for this section

We have  modified the term Chemical in  “Biochemical”

-Total protein units should be added

Thank you for the suggestion: we added the units in the test which is g/dL.

 Sample collection and colony-forming unit fibloblast section

-They are not fibroblast. They have fibroblast morphology. Fibroblastoid colony-forming unit should be better

Yes, it is correct and we have modified it.

 Cellular growth section

-First use of “PD” should be defined

We have defined and explained the formula in the Material and Method session

-In table 2, Decimal numbers are written in an incorrect form

Sure, sorry for the mistake! We have modified the numbers in the table 2

 Differentiation Capacity

 - "HPL-S" and "HPL-E" in the top of the images of figure 6 help the reader

We have added the indications on the top of the figure 6

- Poor resolution in Panel E and F of figure 6 

We changed the figures with a higher resolution

 Material and Methods

 In Result section, 5 donors appear in contrast to the number appears in this section

We modified this part in the Material and method to be clearer

Reviewer 2 Report

1. The new HPL (HPL-S) with the standard (HPL-E) is not common/hard to follow. They doesn’t explain what is S what is E. Also HPL pre

2. Why they not use DMEM and FBS for culture medium like other studies commonly used (Mushahary et al, 2017; https://doi.org/10.1002/cyto.a.23242)? Should they explain in methods 

3. It would be more informative if they also examine the following genes: PPARγ for adipogenesis; runt-related transcription factor-2 (RUNX2), alkaline phosphate (ALP), and osteocalcin (OC) for osteogenesis; and aggrecan (ACAN), collagen type II (COL2b) and SRY-box 9 (SOX9) for chondrogenesis.( Gale et al, 2019; https://doi.org/10.3389/fvets.2019.00178)

4. In conclusion, it’s mentioned about immunomodulatory properties of MSCs but the findings is not present in this paper?

5. The last paragraph in conclusion section doesn’t look like conclusion statement 

“..in cellular therapy with MSCs is now focused on the production of secretome and its clinical use, so in this study, we have also collected the culture medium and isolated the seretome as described in Bari et al [24]. Our aims are to analyse the differences in the secretome isolated from BM-MSCs and expanded in HPL-E and also those in HPL-S to investigate their immunomodulant effects on activated lymphocytes.”

6. Please check the typo and format of the journal before submission, such as Streptomycin or Streptomicine?; CD45-34-14 FITC?; It is mentioned CD45-34-19 in the text page 7 section 2.4, but CD45-34-14 in Fig. 5; I mmunomoulant proprieties?; Results and discussion should be in different section?

7. Please check how to refer book chapter as ref.
a. The cells were counted at optical microscope, using Burker Chamber calculation as indicated in the European Pharmacopoeia (Chapter 2.7.29).

b. Ref 23 is considered as advance online and in press journal articles, unpublished manuscripts and manuscripts submitted for publication, informally published works, or unpublished raw data?

Author Response

Dear reviewer, 

Please find enclosed the revised manuscript entitled “A New Human Platelet Lysate for Mesenchymal Stem Cell Production Compliant with Good Manufacturing Practice Conditions “ 

We wish to thank you for giving us the opportunity to revise and improve our work. 

In this revised version we took into consideration the reviewers’ comments and modified the text accordingly. All alterations in the revised manuscript are tracked using the      "Track Changes" function in Microsoft Word.

We believe that by following the reviewers’ suggestions the new version has been much improved, and hope that it is now suitable for publication in your journal. 

Please, contact us without hesitation      with regard to any further questions or modifications.  

Yours sincerely, 

Katia Mareschi

Below is our response to your comments. 

  1. The new HPL (HPL-S) with the standard (HPL-E) is not common/hard to follow. They doesn’t explain what is S what is E. Also HPL pre

As suggested from the reviewer, we explained the differences in the production of HPL-E and HPL-S and  the meaning of HPL in the Introduction session. We described the composition of HPL in the Material      and Methods section.

  1. Why they not use DMEM and FBS for culture medium like other studies commonly used (Mushahary et al, 2017; https://doi.org/10.1002/cyto.a.23242)? Should they explain in methods 

We have been isolating  MSCs since 1999 following the Material and methods described by Pittenger et al (Science 1999) and they used alpha-MEM + 10 %of FBS. In these years, we have always used  alpha-MEM           testing different additives to ameliorate our isolation  and expansion method also in GMP conditions as described in the references 14 and 15 which have been added also in the paragraph 3.2 

  1. It would be more informative if they also examine the following genes: PPARγ for adipogenesis; runt-related transcription factor-2 (RUNX2), alkaline phosphate (ALP), and osteocalcin (OC) for osteogenesis; and aggrecan (ACAN), collagen type II (COL2b) and SRY-box 9 (SOX9) for chondrogenesis.( Gale et al, 2019; https://doi.org/10.3389/fvets.2019.00178)

We do agree, in principle, with the reviewer that adding more markers for osteogenic, chondrogenic and adipogenic differentiation would further confirm the capacities of the isolated stem cells.

Although, the basal cell characterization for mesenchymal phenotype markers and functional properties performed in our manuscript complies with the      requisite expressed in the guidelines of the International Society for Cell Therapy (ISCT) Ref 7 and 18. According to these guidelines, it is sufficient to demonstrate the ability of MSCs to differentiate into osteoblasts, adipocytes and chondroblasts using standard in vitro tissue culture-differentiating conditions and appropriate cell staining. To this end, differentiation to osteoblasts can be demonstrated by Alizarin Red or von Kossa staining; adipocyte differentiation is most readily demonstrated by staining with Oil Red O; chondroblast differentiation is demonstrated by staining with Alcian blue or immunohistochemical staining of collagen type II.

However, we feel that performing the extensive analysis in the present work, besides the cost in terms of time and money, would not add much to the true message that insists on the validity of the new HPL method for the isolation and purification of E-MSCs under GMP standard for their potential clinical employment. Moreover, for the differentiation and for this method, we used a reduced number of cells while for molecular analysis a higher number is requested. Therefore, in our study design we prefer the staining method to reduce the cost of the experiments that could be redundant.

  1. In conclusion, it’s mentioned about immunomodulatory properties of MSCs but the findings is not present in this paper?

We have just submitted in this journal another manuscript describing the immunomodulating properties of MSCs grown with HPL-E and HPL-S      (Reference 24).

  1. The last paragraph in conclusion section doesn’t look like conclusion statement 

Thanks to the reviewer, we modified the conclusions adding a clearer statement.

  1. Please check the typo and format of the journal before submission, such as Streptomycin or Streptomicine?; CD45-34-14 FITC?; It is mentioned CD45-34-19 in the text page 7 section 2.4, but CD45-34-14 in Fig. 5; I mmunomoulant proprieties?; Results and discussion should be in different section?

We have modified all the typos in the text and an English mother-     tongue teacher has reviewed the whole reviewed manuscript

We reported and discussed the results in the same session following the instruction for the authors for JIMS submission .

  1. Please check how to refer book chapter as ref.
    a. The cells were counted at optical microscope, using Burker Chamber calculation as indicated in the European Pharmacopoeia (Chapter 2.7.29).
  2. Ref 23 is considered as advance online and in press journal articles, unpublished manuscripts and manuscripts submitted for publication, informally published works, or unpublished raw data?

We added the reference as suggested by the review

Our experiment describing the immunomodulant proprieties of the MSCs expanded in the 2 HPLs are described in the ref 24 which is referring to a manuscript, that has been submitted almost simultaneously with this and we are waiting the decision from the editor,

Reviewer 3 Report

In this study, Mareschi et. al. developed a human mesenchymal stem cell (MSC) culture system using human platelet lysate. They also developed a platelet lysate preparation method, termed HPL-S by treating platelet with calcium gluconate. In this method, use of heparin and frequent filtration can be avoided, which makes the method easier and less cumbersome. So, the study looks interesting and important. However, I have a few major concerns, which need to be addressed.

  1. The authors said that in usual method of HPL-E preparation, aggregates are a concern. Hence, frequent filtration of the medium is necessary. Such aggregates did not appear in HPL-S method. This is an important point. A visual representation of the aggregates in the medium, and their microscopic appearance is necessary to show superiority of HPL-S.
  2. The authors said that for equal treatment, medium containing HPL-S and medium containing HPL-E was filtered. However, I think the authors should do experiments without filtering HPL-S containing medium, because the authors claimed that such step can be eliminated by using HPL-S.
  3. Also, as a control, the authors should use standerd method of MSC culture to see the efficacy of their newly developed culture method using HPL-S.

Author Response

Dear reviewer, 

Please find enclosed the revised manuscript entitled “A New Human Platelet Lysate for Mesenchymal Stem Cell Production Compliant with Good Manufacturing Practice Conditions “ 

We wish to thank you for giving us the opportunity to revise and improve our work. 

In this revised version we took into consideration the reviewers’ comments and modified the text accordingly. All alterations in the revised manuscript are tracked using the      "Track Changes" function in Microsoft Word.

We believe that by following the reviewers’ suggestions the new version has been much improved, and hope that it is now suitable for publication in your journal. 

Please, contact us without hesitation      with regard to any further questions or modifications.  

Yours sincerely, 

Katia Mareschi

Below is our response to your comments. 

In this study, Mareschi et. al. developed a human mesenchymal stem cell (MSC) culture system using human platelet lysate. They also developed a platelet lysate preparation method, termed HPL-S by treating platelet with calcium gluconate. In this method, use of heparin and frequent filtration can be avoided, which makes the method easier and less cumbersome. So, the study looks interesting and important. However, I have a few major concerns, which need to be addressed.

  1. The authors said that in usual method of HPL-E preparation, aggregates are a concern. Hence, frequent filtration of the medium is necessary. Such aggregates did not appear in HPL-S method. This is an important point. A visual representation of the aggregates in the medium, and their microscopic appearance is necessary to show superiority of HPL-S.

We thank the reviewer for the thorough and positive assessment of our work. 

We added a figure showing the differences between the HPL before use and between the presence of debris in the culture.

  1. The authors said that for equal treatment, medium containing HPL-S and medium containing HPL-E was filtered. However, I think the authors should do experiments without filtering HPL-S containing medium, because the authors claimed that such step can be eliminated by using HPL-S.

We filtered both mediums      (Alpha MEM + HPL-E and Alpha MEM + HPL-S) when we prepared them before starting the cultures in order to have the same conditions. After the medium was made, the one with HPL-S never needed further filtration, while the one with HPL-E needed to be filtered every time before use. We filtered the medium added with HPL-S only in the first preparation. We have modified the material and method session

  1. Also, as a control, the authors should use standard method of MSC culture to see the efficacy of their newly developed culture method using HPL-S.

In our laboratory, we have validated the use of the standard HPL instead of the FBS and we have already shown that it is a valid replacement (references 14 and 15). Therefore, in this work we compared HPL-S with HPL-E (standard) as the latter is an already validated additive compared to FBS. In this way, we were able to compare two production methods. We have specify it in the first few lines of the Results and Discussion session.

Round 2

Reviewer 3 Report

The authors adequately revised the manuscript. Now the manuscript can be accepted for publication